# Genetics, Pathophysiology, and Current Challenges in Von Hippel–Lindau Disease Therapeutics

**DOI:** 10.3390/diagnostics14171909

**Published:** 2024-08-29

**Authors:** Laura Gómez-Virgilio, Mireya Velazquez-Paniagua, Lucero Cuazozon-Ferrer, Maria-del-Carmen Silva-Lucero, Andres-Ivan Gutierrez-Malacara, Juan-Ramón Padilla-Mendoza, Jessica Borbolla-Vázquez, Job-Alí Díaz-Hernández, Fausto-Alejandro Jiménez-Orozco, Maria-del-Carmen Cardenas-Aguayo

**Affiliations:** 1Laboratory of Cellular Reprogramming, Department of Physiology, Facultad de Medicina, Universidad Nacional Autónoma de México, Av. Universidad No. 3000, Coyoacan CDMX 04510, Mexico; jalim166@gmail.com (L.G.-V.); yetlanetzi@unam.mx (M.V.-P.); lucerocuazozon@gmail.com (L.C.-F.); carmenaguila10@hotmail.com (M.-d.-C.S.-L.); andresivangtz@hotmail.com (A.-I.G.-M.); ibtramon.padilla@outlook.com (J.-R.P.-M.); 2Ingenieria en Biotecnología, Universidad Politécnica de Quintana Roo, Av. Arco Bicentenario, MZ. 11, Lote 1119-33 SM 255, Cancún Quintana Roo 77500, Mexico; jessica.borbolla@upqroo.edu.mx (J.B.-V.); job.diaz@upqroo.edu.mx (J.-A.D.-H.); 3Department of Pharmacology, School of Medicine, Universidad Nacional Autónoma de Mexico, Mexico City 04510, Mexico; alejandrojimenezorozco@gmail.com

**Keywords:** VHL, pVHL, tumor suppressor, HIF, treatments, biomarkers, clinical trials

## Abstract

This review article focuses on von Hippel–Lindau (VHL) disease, a rare genetic disorder characterized by the development of tumors and cysts throughout the body. It discusses the following aspects of the disease. Genetics: VHL disease is caused by mutations in the VHL tumor suppressor gene located on chromosome 3. These mutations can be inherited or occur spontaneously. This article details the different types of mutations and their associated clinical features. Pathophysiology: The underlying cause of VHL disease is the loss of function of the VHL protein (pVHL). This protein normally regulates hypoxia-inducible factors (HIFs), which are involved in cell growth and survival. When pVHL is dysfunctional, HIF levels become elevated, leading to uncontrolled cell growth and tumor formation. Clinical Manifestations: VHL disease can affect various organs, including the brain, spinal cord, retina, kidneys, pancreas, and adrenal glands. Symptoms depend on the location and size of the tumors. Diagnosis: Diagnosis of VHL disease involves a combination of clinical criteria, imaging studies, and genetic testing. Treatment: Treatment options for VHL disease depend on the type and location of the tumors. Surgery is the mainstay of treatment, but other options like radiation therapy may also be used. Challenges: This article highlights the challenges in VHL disease management, including the lack of effective therapies for some tumor types and the need for better methods to monitor disease progression. In conclusion, we emphasize the importance of ongoing research to develop new and improved treatments for VHL disease.

## 1. Introduction

von Hippel–Lindau (VHL) disease is a rare, autosomal dominant genetic disorder characterized by the development of multiple tumors and cysts throughout the body, resulting from germline mutations or deletions in the VHL tumor suppressor gene located on chromosome three [1]. VHL affects approximately 1 in 36,000 individuals globally [2]. While the condition can start at any age, the median age of onset is 26 years, with a high penetrance of disease manifestations by age 65 for 97% of the cases [3].

The clinical spectrum of VHL is broad, encompassing benign and malignant tumors involving the central nervous system, retina, kidneys, adrenal glands, pancreas, and other organs [4]. Hemangioblastomas, highly vascular tumors that can arise in the central nervous system, retina, and other locations, are a hallmark of VHL [5]. While most VHL-associated tumors are benign, they can cause significant morbidity due to their location and size. Notably, VHL patients are at increased risk for developing clear cell renal cell carcinoma and pheochromocytoma [6,7].

The underlying pathophysiology of VHL involves dysregulation of the hypoxia-inducible factor (HIF) pathway [7,8]. The VHL protein is a critical component of the E3 ubiquitin ligase complex that targets HIF for degradation under normoxic conditions. In the absence of functional VHL, HIF is stabilized, leading to aberrant activation of genes involved in angiogenesis, erythropoiesis, and glucose metabolism [3,9].

Differential diagnosis of VHL can be challenging due to the overlap of clinical features with other genetic and sporadic conditions. To establish a definitive diagnosis, genetic testing to identify pathogenic variants in the VHL gene is essential [10,11,12,13] (Richards et al., 2015).

## 2. Von Hippel–Lindau (VHL) Disease: Highlights

VHL disease is a multiorgan neoplastic syndrome with autosomal dominant transmission, complete penetrance, and variable expression caused by mutations in the *VHL* gene. Although VHL disease is hereditary in many cases, new mutations cause up to 20% of the cases [14]. Pathogenic variants in the *VHL* gene predispose individuals to tumors and cysts in many organ systems. These include brain and spinal cord hemangioblastoma, renal cell carcinoma (RCC), retinal hemangioblastoma (RH), pheochromocytoma, epididymal and broad ligament cystadenomas, endolymphatic sac tumor, pancreatic neuroendocrine tumors, and renal and pancreatic cysts [4,15].

### 2.1. History

Observations relating to VHL disease first appeared in the 19th century. For example, in 1879, Panas and Rémy illustrated a retinal hemangioblastoma for the first time. Subsequently, von Hippel contributed to the description of the disease through clinical data obtained from a patient with multiple retinal lesions that appeared over several years. From all her observations, von Hippel concluded that the primary retinal lesion was a hemangioblastoma. Later, in 1926, Lindau published a monograph. In this document, he brought together into one coherent entity the retinal, cerebral, and visceral components of this disease. Lindau designated “central nervous system angiomatosis” as this disease entity because he believed that visceral concomitants did not manifest by symptoms. Finally, ‘Lindau’s disease’ was defined as an association of cerebellar hemangioblastoma with one or more lesions: retinal hemangioblastoma (the “von Hippel tumor”), spinal cord hemangioblastoma, pancreatic cysts, renal and epididymal abnormalities, and the existence of at least one other family member with the disease [16]. This term changed to the ‘von Hippel–Lindau (VHL)’ disease in the 1970s. In 1988, a report explained the linkage of the *VHL* gene to chromosome 3 [17], and, in 1993, a research group identified the VHL tumor suppressor gene [1].

### 2.2. Etiology

von Hippel–Lindau (VHL) syndrome is a syndrome associated with functional inactivation of the von Hippel–Lindau protein (pVHL) [18]. The von Hippel–Lindau protein (pVHL) is a tumor suppressor mainly known for its role as a regulator of hypoxia-inducible factor (HIF) activity [19]. The homonymous *VHL* gene codifies the pVHL protein, localizes on chromosome 3p25, and is expressed in fetal and adult tissues [20]. On the other hand, VHL-related tumor development follows Knudson’s ‘two-hit model’ of tumorigenesis [21]: Patients with VHL are born with a germline mutation in one copy of their VHL gene in all cells (first hit or event). Somatic mutation in the other copy of the *VHL* gene (second hit or event) initiates tumor development in a particular cell [22,23].

### 2.3. Epidemiology

The disease reports penetrance of > 80% by age 60 and an approaching 100% by age 75. The prevalence of VHL disease is between 1 in 39,000 and 1 in 91,000 individuals in different regional populations [24,25]. Most are >20 years old and are prone to selection bias due to the inclusion of clinically affected patients with VHL disease diagnosed before genetic testing was available. For example, VHL disease had a birth incidence of 1 in 36,000 live births in eastern England, and prevalences of 1 in 39,000 individuals in southwestern Germany and 1 in 53,000 individuals in eastern England [26,27]. However, in Denmark, in an unselected cohort of all known Danish carriers of a disease-causing variant, VHL prevalence is estimated at 1 in 46,900 individuals and an incidence of 1 in 27,300 live births [28].

### 2.4. Pathophysiology

von Hippel–Lindau (VHL) syndrome is a rare hereditary cancer characterized by the development of benign or malignant tumors in specific topographic locations. In this sense, central nervous system hemangioblastoma and clear cell renal cell carcinoma (RCC) are the most frequently tumors [4].

Tumorigenesis in VHL syndrome is linked to the loss of function of the VHL tumor suppressor protein in cell differentiation [29] where hypoxia-inducible factors (HIF1 and HIF2) are activated and accumulate in the cell [30,31]. The consequences of this up-regulation include transcriptional activation of genes containing hypoxia-responsive elements [32,33]. However, it remains unclear why the loss of VHL function and subsequent HIF activation lead to tumorigenesis. Tumorigenesis is associated with the PI3K/Akt signaling pathway, and this pathway is also associated with HIF activation and deactivation of tumor suppressors. Further research is needed in this area associated with VHL [34]. The PI3K/AKT signaling pathway is a critical regulator of various cellular processes, including metabolism, growth, and survival; it has a significant role in cancer biology, and is associated with HIF activation and the deactivation of tumor suppressors. 

HIF Activation: The PI3K/AKT pathway can activate hypoxia-inducible factor 1 alpha (HIF1α) even under normoxic conditions (normal oxygen levels) [34]. This occurs through several mechanisms. Increased Stability: PI3K/AKT signaling can inhibit the prolyl hydroxylases (PHDs) that normally target HIF1α for degradation. When PHDs are inhibited, HIF1α becomes more stable and accumulates in the cell, leading to its activation and subsequent transcription of target genes involved in angiogenesis, metabolism, and cell survival [35]. Transcriptional Regulation: Activated AKT can also enhance the transcription of HIF1α by promoting its translation and increasing its mRNA levels, further contributing to its accumulation and activity in cancer cells [36]. Deactivation of Tumor Suppressors: The PI3K/AKT pathway is known to negatively regulate several tumor suppressor proteins, which can lead to enhanced tumorigenesis. PTEN (Phosphatase and Tensin Homolog): PTEN is a well-known tumor suppressor that negatively regulates the PI3K/AKT pathway. Loss of PTEN function leads to increased AKT activity, promoting cell survival and proliferation while inhibiting apoptosis. This loss of PTEN is common in various cancers, including breast cancer [37]. p53: The tumor suppressor p53, which is crucial for DNA repair and apoptosis, can also be inhibited by the PI3K/AKT pathway. AKT can phosphorylate and inactivate p53, leading to reduced apoptosis and enhanced cell survival, contributing to cancer progression [38]. FoxO Transcription Factors: AKT can phosphorylate FoxO transcription factors, leading to their exclusion from the nucleus and subsequent degradation. FoxOs are involved in the regulation of genes that promote cell cycle arrest and apoptosis, so their inactivation by AKT promotes tumor growth and survival [39]. In summary, the PI3K/AKT signaling pathway promotes the activation of HIF1α, which supports tumor growth and survival, while simultaneously deactivating key tumor suppressors like PTEN and p53, facilitating cancer progression and resistance to therapies.

Another explanation of why the loss of VHL function and subsequent HIF activation lead to tumorigenesis suggests that the ‘second hit’ would cause loss of pivotal VHL function during organ development, leading to maldeveloped structures that represent prerequisites for tumor formation [40].

## 3. Genetics

### 3.1. VHL Gene and Protein

In 1988, Seizinger and colleagues mapped the *VHL* gene. The findings showed that the *VHL* gene is located on the short arm of chromosome 3 in the 3p 25-25 region (Figure 1). Moreover, the *VHL* gene has 14,500 base pairs of genomic DNA, and 852 nucleotides with three exons that can generate two mRNAs through a process that causes the loss of exon 2. This gene is conserved in rodents, flies, and worms [41,42].

The *VHL* gene consists of three exons with the capacity to generate two mRNA transcripts. The mRNA1 transcript is composed of exons 1, 2, and 3, and the mRNA2 transcript is composed of exons 1 and 3 (Figure 2). The mRNA1 transcript encodes two pVHL proteins. When translation occurs, the complete protein from 1 to 213 amino acids with a molecular mass of 24~30 kDa, also known as pVHL30, and a small peptide from 54 to 213 amino acids is translated through an alternative start codon at codon 54 with a molecular mass of 18~19 kDa, also known as pVHL19. Both proteins (pVHL30 and pVHL19) are localized in the cytoplasm and cell nucleus, respectively [43,44].

The structure of the pVHL protein has two domains: the β domain that ranges from 63 to 155 aa and the α domain that includes amino acids 156 to 193, as shown in Figure 3. On the other hand, the protein pVHL30 contains an acidic domain located in amino acids 1–54, absent from the pVHL19 protein. The function of pVHL30 is present in several events. For example, protein degradation via the proteasome is the most studied, and in addition to both isoforms, it appears to maintain tumor suppression activity [45].

### 3.2. Inheritance

VHL disease is autosomal dominant; that is, it is transmitted from parents to children. However, the disease can occur sporadically. The mutation is inherited when one parent has the *VHL* gene altered (mutated), so each child has a 50% chance of inheriting it. Consistent with the dominant inheritance pattern, it is enough to have an altered gene to develop the disease. Thus, an affected parent (who therefore has a mutated gene and a normal one) could pass on the mutated gene to his offspring, who in turn will develop the disease despite having the other normal gene (see Figure 4) [46]. VHL disease is also autosomal, which means that men and women can suffer from it equally (Figure 4). People with parents, brothers, or sisters with VHL have a 50% risk of disease. If the person has an uncle, cousin, or grandparent with VHL, they are also at risk for developing VHL. Moreover, the presence of the disease in individuals without family background has been reported, therefore it is necessarily a differential diagnosis for each individual [47].

### 3.3. Mutations

In *VHL* gene mutation carriers, the development of the disease begins with the loss or inactivation of the wild-type allele. Cytogenetic abnormalities, mutations, or hypermethylation cause inactivation of the remaining ‘healthy’ allele (Figure 5). Currently, it is unknown why the second event occurs in some tissues and not in others [48]. On the other hand, some studies show that patients develop retinal hemangioblastoma with complete deletion of the *VHL* gene less frequently than those with a single amino acid substitution in the *VHL* gene. In addition, there were no differences between patients with a truncated variant and those with a single amino acid substitution [49,50].

Germline VHL mutations cause different protein defects that result in the clinical heterogeneity of VHL disease. For example, type 1 VHL disease (without pheochromocytoma) is associated with mutations that cause the complete unraveling of the protein structure (missense mutations in the hydrophobic core of the VHL protein, protein-truncating mutations, and partial gene deletions). On the other hand, VHL disease type 2 (with phaeochromocytoma) is associated with missense mutations at pVHL protein binding sites, causing local defects [51]. Table 1 shows several examples of the relationship between the type of VHL disease and some mutations.

Furthermore, Nordstrom-O’Brien reports a mutation spectrum of 43.2% in exon 1, 17% in exon 2, and 39.8% in exon 3 [65]. Mutations are heterogeneous and are widely distributed throughout the coding sequence, especially in exons 1 and 3. Missense mutations are the most common type of mutation (61%), followed by frameshift (15.7%), nonsense (13.2%), in-frame insertions/deletions (6.6%), and splicing mutations (3.5%) [90]. Finally, wild-type allele inactivation may arise due to allelic loss, hypermethylation, or point mutations [91,92].

pVHL has two domains: domain alpha and domain beta. Most pathogenic missense mutations are found in two regions of the VHL protein that interact with the elongin C-binding site. In line with this, several tumor mutations locate in alpha-domain, in the amino acids that contact with elongin C (residues 158–184 of the α domain) [93]. The rest of the mutations are in the beta domain (residue 65–117 of the β domain) [82,94]. Reports indicate that mutations in Ser111 and Trp117 of pVHL block HIF binding [95]. Most recurrent mutations are the result of de novo mutations in mutation-susceptible regions, known as hot spots. The most common hot spots are delPhe76, Asn78Ser, His, Thr, Pro86Leu, Arg 161 Stop, Cys 162Tyr, Arg167Gln, Trp, and Leu78Pro [54]. Fourteen mutations (5.5%) locate at Ser65, nine mutations (3.5%) at Trp117, (3.1%) at Phe76, (2.8% each) at Asn78, Ser80, Leu135, and Arg161, (2.4%) at His115, and five mutations (2% each) at Gly114 and Leu184 [96]. Common germline mutations in VHL are delPhe76, Asn78Ser, Argl61Stop, Arg167Gln, Argl67Trp, and Leu178Pro [44,54]. Moreover, it has been elucidated the effect of VHL Arg200Trp mutation on HIF-1 interaction and ubiquitination [97]. In line with this, specific missense mutations (Arg167Trp and Arg167Gln) are associated with a high risk (62%) of phaeochromocytoma [51]. Furthermore, amino acid substitution (Glu70Lys) at the HIF-α binding site increments specific risk of developing CNS hemangioblastoma [82], while missense mutations (Leu118Pro or Arg167Trp) are associated with renal cancer [98] (Figure 6). Finally, the Try98His and Asn (Asn78Ser) associate with a low risk of RCC (type 2A phenotype).

## 4. Manifestations, Diagnosis, and Treatment of VHL Disease

Symptoms of VHL disease vary among patients and depend on the size and location of the tumors. On the other hand, diagnosis can be made based on specific clinical criteria (signs, symptoms, and imaging), or when molecular genetic testing reveals a change in the *VHL* gene. Finally, treatment depends on the location and size of the tumors and usually involves surgical removal of the tumors. Radiation therapy may be used in some cases (Table 2).

## 5. Molecular Basis of VHL Disease

The best known pVHL function is the regulation of hypoxia-inducible factor-alpha (HIF-α) protein levels through degradation under normoxic conditions [115]. The interaction between pVHL and HIF-α requires the hydroxylation dependent on prolyl-4 hydroxylase domain enzymes (PHD1, -2 and -3) of at least one of two specific proline residues of HIF-α [116]. Subsequently, HIF-α is ubiquitylated and degraded via the proteasome. Under hypoxic conditions or in the presence of mutant pVHL [60,117], the pVHL complex cannot recognize HIF-α, then HIF-α accumulates in the cytoplasm. In the cytoplasm, HIF-α forms a heterodimer with HIF-β. Subsequently, the heterodimer translocates to the nucleus with the transcriptional coactivator p300, where it binds to response elements (HRE) [118] and promotes the transcription of many genes involved in angiogenesis, glucose metabolism, cell survival, and tumor progression [9] (Figure 7).

However, the pathogenesis of clear cell renal cell carcinomas (ccRCCs) implies mTOR complex 1 (mTORC1). For example, reports show that mTORC1 activates in 60% to 85% of ccRCCs [119].

mTOR is a serine/threonine protein kinase. mTOR nucleates two different complexes, mTORC1 and mTORC2 [120]. mTORC1 is composed of mTOR, the regulatory associated protein of mTOR (RAPTOR), and the protein mammalian lethal with sec-teen protein 8 [121]. Some mTORC1 substrates are S6 kinase 1 (S6K1) and the eukaryotic initiation factor 4E (eIF4E) binding protein 1 (4E-BP1). Finally, phosphorylation of S6K1 and 4E-BP1 by mTORC1 stimulates protein translation [122]. Targeting the PI3K/Akt/mTOR pathways is a promising strategy in cancer, and possible in VHL treatment due to their central role in tumor biology, potential to overcome resistance, and ability to enhance the effectiveness of existing therapies [123].

In response to hypoxia, the regulation of mTORC1 involves the protein regulated in development and the DNA damage response 1 (REDD1). In this condition, HIF binds to a response element in the REDD1 promoter for induction, and therefore negatively regulates mTORC1 [124] (Figure 7a right). Paradoxically, mTORC1 is broadly activated in ccRCCs, pVHL-inactivated, HIF-activated, and up-regulated REDD1. These findings suggest that other mechanisms favor tumors to escape growth suppressor signals resulting from pVHL loss and up-regulation of REDD1 [125] (Figure 7b left).

Another HIF-dependent mechanism that activates mTORC1 is the down-regulation of the mTOR inhibitor, the DEP domain-containing mTOR-interacting protein (DEPTOR). DEPTOR is significantly down-regulated in pVHL-deficient ccRCC tumors and cell lines. In this tumor type, DEPTOR is transcriptionally suppressed by both HIF-1 and HIF-2, mediated by the HIF target gene, BHLHe40 [126] (Figure 7b left).

Finally, a study revealed a new mechanism for the deregulation of mTORC1 in ccRCC. The report showed that pVHL represses the regulatory-associated protein of mTOR (RAPTOR), inhibiting mTORC1 signaling. Therefore, the loss of pVHL function in ccRCC is consistent with the hyperactivation of mTORC1 signaling. This mechanism describes a novel pVHL-mediated regulation of mTORC1 by targeted ubiquitination and degradation independent of HIF [127] (Figure 7b right).

pVHL is a critical tumor suppressor in ccRCCs. On the other hand, most patients with ccRCC are drug-resistant to therapies. Although targeted therapies that inhibit angiogenesis and mTOR pathways can lead to initial tumor control, most patients develop resistance [128,129,130]. Therefore, the identification of additional pVHL substrates could improve therapeutic options for ccRCC. Next, we review some pVHL substrates and possible therapeutic targets in ccRCCs.

Most reports on pVHL/HIF transcriptional activation have focused on HIF-bound promoters. However, evidence suggests distal enhancer elements in pVHL/HIF transcriptional control. In this sense, a study identified a master regulator crucial for the pathogenesis of ccRCC, ZNF395. pVHL loss stabilizes HIF2α occupancy in tumor-specific gained enhancers. HIF2α recruits histone acetyltransferase p300 to maintain H3K27 acetylation, up-regulating the expression of ccRCC-specific genes such as ZNF395. ZNF395 has a functional role in ccRCC tumorigenesis in vitro and in vivo [131] (Figure 8a-1).

A genome-wide in vitro expression strategy to identify proteins that bind to pVHL when hydroxylated determined that zinc fingers and homeoboxes 2 (ZHX2) are a novel pVHL substrate transcription factor. Analysis of tumors from ccRCC patients and pVHL loss-of-function mutations confirmed pVHL loss usually increases the abundance and nuclear levels of ZHX2 in ccRCC tumors. Mechanically, this study revealed that ZHX2 could promote NF-kB activation and carcinogenesis of ccRCC [132]. Another study showed the downstream signaling pathway of ZHX2. ZHX2 facilitated proliferation and migration in ccRCC cell lines by activating the MEK1/ERK1/2 signaling pathway. Furthermore, ZHX2 overexpression could induce Sunitinib resistance by activating autophagy through the MAPK/ERK signal pathway [133] (Figure 8a-2 up).

Another novel genome-wide in vitro expression strategy coupled with a GST-binding screen for pVHL substrates identified Scm-like with four malignant brain tumor domains 1 (SFMBT1) as a direct target of pVHL. In renal tumors compared to adjacent normal tissue, the levels of SFMBT1 are high. On the other hand, the functional characterization of SFMBT1 showed that it promotes ccRCC cell proliferation, anchorage-independent growth, and tumor xenograft growth. The analysis also identified sphingosine kinase 1 (SPHK1). SPHK1 is an SFMBT1 target gene that contributes to the oncogenic phenotype of renal tumors [134]. SFMBT1 and its downstream target gene SPHK1 could represent a new therapeutic strategy for patients with ccRCC treatment (Figure 8a-2 below).

Finally, mRNA expression levels of both transcription factors (ZHX2 and SFMBT1) in a tissue microarray constructed using 97 ccRCC samples showed an association with overall survival (OS) and disease-free survival (DFS) analyses. In this sense, survival analysis demonstrated that poor clinical outcomes in patients with ccRCC were associated with combined high expression levels of SFBMT1 and ZHX2. These results suggest that the co-expression of these two targets could be a promising biomarker for predicting the outcome of patients with ccRCC [135].

However, a study describes the pathologic angiogenic phenotype of pVHL missense mutations in ccRCC. In this case, the SUMO Enhancer (RSUME) sumoylates pVHL mutants. This post-translational modification promotes HIF-2α stabilization and leads to enhanced VEGF action. Therefore, it promotes more vascularized tumors. Regarding pathology, RSUME levels are higher in tumor patients with pVHL mutations and are associated with a poor prognosis in RCC tumors. These findings suggest that RSUME could be a biomarker of the outcomes of renal cell carcinoma [136] (Figure 8a-3).

Finally, tumors from ccRCC patients with pVHL loss show elevated TANK binding kinase 1 (TBK1). A study discovered that loss of pVHL or hypoxia in cancer hyperactivates TBK1. As a result of TBK1 hyperactivation, TBK1 phosphorylates p62 on the Ser366 residue and promotes renal tumorigenesis [137] (Figure 8a-4).

Regarding SNC hemangioblastomas, the analysis of tissues with mutations in the expressions of *VHL* gene showed the JAK2 and STAT3. The results of this study indicate that pVHL binds to JAK2 and STAT3 and mediates its ubiquitination. Furthermore, the study suggests that hemangioblast progenitor cells can differentiate into neoplastic cells by activating the JAK-STAT signaling pathway [138] (Figure 8b-1).

## 6. Animal Models for the Study of VHL

Current animal models for VHL disease can partially capture the disease by showing the involvement of a particular organ; therefore, much work is still needed to develop a model that exhibits several of the clinical manifestations (Table 3). Other manifestations of VHL disease, such as tumors of the endolymphatic sac, middle ear, pheochromocytoma, cerebellar, and cervical hemangioblastomas, are not seen in almost any of the models. Additionally, phenotypes not associated with VHL disease can also arise from the product of the mutated VHL protein [139,140].

The first animal model for VHL disease led to the development of an aberrant placenta, and thus was lethal to the embryo. Mice died in utero at embryonic days 10.5 and 12.5 [141]. However, the use of conditional VHL knockouts in mice has been shown to be an effective approach to delineate the role of VHL in individual organ systems.

Eyes. To date, there is still no adequate model to study the CNS and retinal hemangioblastomas. A zebrafish model expressing retinal neovascularization from vascular leakage, edema, and retinal detachment has been developed [142]. Studies have shown an increase in VEGF and CXCR4A in the CNS in these zebrafish. In fact, this model manifests certain aspects of age-related macular degeneration, diabetic retinopathy, and some cases of VHL [143]. However, zebrafish do not develop hemangioblastomas; van Rooijen and colleagues were able to use this zebrafish model to demonstrate inhibition of angiogenesis through the administration of VEGF receptor tyrosine kinase inhibition.

Kidney. CCRCC is a malignant kidney neoplasm that arises sporadically or is inherited through inactivation of VHL. Rankin et al. were the first group to successfully create a model that generated renal microcysts and macrocysts with similar morphological and molecular characteristics found in VHL-associated kidney disease [144]. Using Cre-loxP, they eliminated VHL expression from the proximal tubule using the phosphoenolpyruvate carboxykinase (PEPCK) promoter to drive Cre expression. Other groups have managed to obtain phenotypes that manifest acute nephritis with hematuria, proteinuria, and renal failure; characteristic features of pauci-immune RPGN (glomerulonephritis crescent) with prominent segmental fibrin deposition and fibrinoid necrosis [145].

Pancreas. Shen et al. produced a model using the insulin promoter factor 1 (Pdx) promoter to drive Cre expression. Survivors secondary to incomplete penetrance expressed highly vascularized cysts and microcystic adenomas, eliminating VHL expression throughout the pancreas [146].

Reproductive system. Ksp1.3-Cre mice were crossed with Vhlh^fl^/^fl^ mice and Pten^fl^/^fl^ mice. The modified mice were then bred to generate a mouse model with dual VHL and Pten deficiencies specific for genital tract epithelium. These mice were able to recapitulate clear cell cystadenoma of the genital tract. in both males and females [147].

Liver. A mouse model in which VHL was acutely inactivated in utero exhibited embryonic lethality with liver necrosis and vascular defects [148]. A mouse model that conditionally inactivated VHL in hepatocytes led to hepatic hemangiomas [149]. Similarly, another model that conditionally inactivated VHL in a mosaic pattern in multiple organs showed liver hemangiomas, as well as angiectasias in the pancreas, heart, lung, and kidney [150]. Some other models showed growth deficiency, angiectasias, hemangiomas, endothelial cell proliferation, severe liver steatosis (accumulation of neutral fat in hepatocytes), and inflammatory cell infiltration [144].

In recent years, various animal models have been proposed in which the main objective is the inactivation of the *VHL* gene product in various organs (Table 3). In these models, the mechanisms associated with HIF and its link with tumorigenesis. However, these models are generally considered incapable of recapitulation of the most common characteristics of human VHL disease. To date, there are no models that develop retinal hemangioblastomas, the most common clinical manifestation of the disease [151].

## 7. Biomarkers in VHL Disease

Biomarkers for VHL disease are scarce. However, some studies propose several. For example, monitoring plasma levels of HIF-dependent molecules would allow monitoring of disease activity in VHL patients [101]. Although a report confirmed that elevated plasma VEGF levels are associated with an increased risk of dying from ccRCC, plasma VEGF levels are also significantly increased in tumors without VHL alteration. It suggests that VHL-independent mechanisms are involved in up-regulation of VEGF in ccRCC [152]. On the other hand, in serum from patients with VHL, no correlation was found between VEGF levels and the presence of manifestations of VHL disease [153]. Finally, in another study, the presence of abnormalities in VHL did not correlate with overall survival (OS), disease-specific survival (DSS), and progression or recurrence-free survival (PFS), and the expression of VEGF had no prognostic value. However, this study showed an association with a poorer prognosis in patients with no expression of VHL and HIF1-α expression and patients with overexpression of ERK5 [154].

The remarkable phenotypic heterogeneity in organ involvement and tumor onset age between and within VHL families has not allowed reliable markers to predict the age-related tumor risks in VHL patients. In this sense, the information shown in Table 4 is a compilation of several molecules proposed as biomarkers. The samples are mainly renal tissues. The recruited subjects are patients with VHL disease with ccRCC, while some patients have VHL disease with hemangioblastoma or pancreatic lesions. The heterogeneity of the disease concerning clinical and molecular issues manifests itself in the diversity of the molecules described in the table. Last, several molecules are prognostic biomarkers, which means those that indicate the probability of a change in a future clinical event, disease recurrence, or progression in an identified population [155].

The main limitation of these studies is the relatively small number of patients, but they provide an approximation to standard clinical and pathological data that are still essential in the development of biomarker panels for von Hippel–Lindau disease in addition to the molecular mechanisms that underlie this disease. Table 4 resumes the current principal biomarkers for VHL as follows: A study involving 300 patients with von Hippel–Lindau (VHL) disease and 92 healthy family controls found that VHL patients exhibited significantly shorter telomere lengths, which were associated with increased risks of various VHL-related tumors, indicating that telomere length could serve as a biomarker for VHL risk [156]. In metastatic renal cell carcinoma (mRCC) patients, tumors with double c-Myc/HIF-2α-positive staining were linked to worse outcomes with sunitinib treatment [157]. SERPINH1 was identified as a potential prognostic marker for disease-free survival (DFS) in patients with clear cell renal cell carcinoma (ccRCC) [158]. Positive PD-L1 expression in ccRCC correlated with more aggressive disease in both sporadic and hereditary VHL-associated cases [159]. High lamin B1 expression was found to be an unfavorable prognostic marker in primary RCC [160]. Low expression of CTLA-4 and PD-1, along with a low fraction of immune cells, was associated with poor survival in high-risk kidney cancer patients [161]. Reduced expression of lncRNA FGD5-AS1 in ccRCC was linked to shorter overall survival (OS) and DFS [162]. High levels of SFMBT1 and ZHX2 were associated with poor outcomes in ccRCC [135]. In imaging studies, 89Zr-bevacizumab PET showed potential in visualizing VHL disease manifestations, though it could not predict lesion behavior [163], and 68Gallium-DOTATATE PET/CT demonstrated higher detection rates in pancreatic lesions associated with VHL compared to CT and MRI [164].

## 8. Effect of Regional Populations on VHL Disease

While the genetic basis of VHL disease remains consistent across populations, the prevalence and specific manifestations can vary regionally. Several factors contribute to these differences:

Founder effects: In isolated populations, specific VHL gene mutations may be more prevalent due to a shared common ancestor. This can lead to variations in disease phenotype and severity [165]. 

Genetic background: Different genetic backgrounds can modify the expression and penetrance of VHL disease. Polymorphisms in genes interacting with VHL or genes involved in tumorigenesis pathways may influence disease progression [166]. 

Environmental factors: Exposure to certain environmental factors, such as diet, lifestyle, and toxins, may interact with genetic susceptibility to VHL disease, leading to variations in disease presentation [167]. 

Healthcare access and practices: Differences in healthcare access and practices can impact the early detection, diagnosis, and management of VHL disease, leading to variations in disease outcomes [168]. 

It is essential to consider these regional differences when studying VHL disease to develop targeted prevention, early detection, and treatment strategies.

## 9. Targeted Therapy in VHL Disease

Clinical research on VHL disease is summarized in the next table (Table 5) As follows: A series of clinical trials investigating various therapeutic agents for conditions related to von Hippel–Lindau (VHL) disease and renal cell carcinoma (RCC) yielded mixed outcomes. Sunitinib trials (NCT00673816 and NCT00330564) were terminated due to slow recruitment and adverse events, with some limited efficacy in RCC lesions [169,170]. The combination of ranibizumab and E10030 (NCT02859441) showed a limited effect on VHL-associated retinal hemangiomas but maintained a reasonable safety profile [171]. Neovastat (Protocol CT/AE-941/002 and III) demonstrated a survival benefit and good tolerability in RCC patients, especially in a higher dose group [172,173]. Propranolol (EudraCT 2014-003671-30) showed promise in stabilizing retinal hemangioblastomas in VHL patients with minimal adverse effects [174]. However, sorafenib (EudraCT 2007-002132-29) showed no response in CNS hemangioblastomas over six months [175]. Pazopanib (NCT01436227) and belzutifan (NCT03401788) were associated with tumor size reduction and disease stabilization in VHL-related conditions [176,177,178]. Other trials, such as those involving PT2385, dovitinib, ranibizumab, vorinostat, and vandetanib, showed varying degrees of efficacy and tolerability, with some requiring further exploration to optimize their clinical benefits [179,180,181]. Finally, tyrosine kinase inhibitors (TKIs) demonstrated partial responses in VHL-related tumors, particularly RCC and pancreatic lesions, with acceptable side effects in a study by the Medical Ethics Committee of Peking University [182].

## 10. Discussion

The current review provides a comprehensive overview of von Hippel–Lindau (VHL) disease, a rare genetic disorder characterized by multiple organ neoplastic syndrome. The disease is caused by deletions or mutations in the *VHL* gene [183], resulting in the development of cysts and tumors in various organs, including the brain, spine, eyes, kidneys, pancreas, adrenal glands, inner ears, reproductive tract, liver, and lungs [108]. The prevalence of VHL disease is estimated to be between 1 in 39,000 and 1 in 91,000 individuals in different regional populations [24,25]. The penetrance of the disease is high, with a significant risk of developing clinical manifestations throughout life, emphasizing the importance of thoroughly understanding the underlying genetics and pathophysiology. Here, we discuss the etiology, epidemiology, pathophysiology, genetics, clinical manifestations, diagnosis, and current treatments, as well as the molecular aspects of the disease. *VHL* is inherited in an autosomal dominant manner and is associated with mutations in the *VHL* gene. It discusses the role of von Hippel–Lindau protein (pVHL) as a tumor suppressor known for regulating hypoxia-inducible factor (HIF) activity [19]. We provide insights into the inheritance pattern, mutations, and animal models used to study this disease, emphasizing the need for more comprehensive models that capture the diverse clinical manifestations of the disease. Furthermore, this review explores the biomarkers for VHL disease (Table 4). We discuss the proposed biomarkers for VHL disease that include monitoring plasma levels of HIF-dependent molecules, such as VEGF, which may allow for the monitoring of disease activity in VHL patients. However, it is important to note that elevated plasma VEGF levels are also significantly increased in tumors without VHL alteration, suggesting the involvement of VHL-independent mechanisms in up-regulating VEGF in clear cell renal cell carcinoma (ccRCC) [101,152,153,184]. Additionally, studies have shown that the presence of abnormalities in VHL does not correlate with overall survival, disease-specific survival, and progression or recurrence-free survival. Telomere length is another potential biomarker for risk assessment in VHL patients. Shorter blood telomers are reported in VHL patients [156]. The limitations of using blood telomere length as a biomarker for VHL disease include the relatively small number of patients studied, which may impact the generalizability of the findings. Additionally, the heterogeneity of the disease in terms of clinical and molecular aspects may affect the reliability of telomere length as a predictive marker for age-related tumor risks in VHL patients. Furthermore, it is argued that while shorter blood telomere length is associated with higher age-related risks of VHL-associated central nervous system hemangioblastomas, renal cell carcinoma, pancreatic cysts, and neuroendocrine tumors, the correlation may not be consistent across all patients with VHL disease. Therefore, the use of blood telomere length as a biomarker for VHL disease may have limitations in accurately predicting tumor risks and disease progression in all individuals with VHL.

Therefore, while these biomarkers show promise, further research is needed to establish their reliability and clinical utility in predicting tumor risks and disease progression in VHL patients.

The identification of mutations in the *VHL* gene has been crucial for the diagnosis and management of VHL [185]. Mutations in this gene have been observed to predispose affected individuals to develop a variety of tumors. Understanding how these mutations lead to tumor formation is critical for the development of more specific and effective therapeutic approaches.

In terms of pathophysiology, it has been demonstrated that the inactivation of the wild-type allele of the *VHL* gene is an initial event in the development of the disease, followed by the loss of function of the second allele, triggering tumorigenesis. This “two-hit” model proposed by Knudson has been fundamental in understanding tumor progression in VHL and has led to deeper investigations into the molecular mechanisms involved in this disease.

Furthermore, the review highlights the importance of clinical heterogeneity in VHL, with significant variations in disease presentation even within the same family. This underscores the need for an individualized approach in the diagnosis and management of patients with VHL, considering both genetic and clinical aspects.

The current review discusses the various manifestations of VHL disease, such as retinal hemangioblastomas, renal cell carcinomas, pheochromocytomas, and pancreatic cysts, and the corresponding diagnostic modalities and treatments. Furthermore, it sheds light on the molecular basis of VHL disease, elucidating the role of the pVHL protein in regulating hypoxia-inducible factor-alpha (HIF-α) protein levels through degradation under normoxic conditions. It also explores the association between VHL loss and the activation of the mTOR pathway [119], highlighting the significance of various pVHL substrates, such as ZNF395, ZHX2, SFMBT1, RSUME, and TBK1, in the pathogenesis of clear cell renal cell carcinomas [136,137,138].

Ongoing research in the genetics and pathophysiology of von Hippel–Lindau disease is crucial for improving early diagnosis, clinical management, and the development of more effective therapies. Understanding the underlying molecular mechanisms of VHL will not only expand our knowledge of this disease but also open up new opportunities for more precise and personalized therapeutic interventions in the future.

## 11. Conclusions

The genetic basis of VHL disease, characterized by mutations in the *VHL* gene, plays a central role in predisposing individuals to a spectrum of tumors and cysts in various organs. Further research into the genetic mechanisms underlying VHL is essential for advancing diagnostic and therapeutic strategies.

The pathophysiology of VHL syndrome, involving the loss of function of the VHL tumor suppressor protein and subsequent activation of hypoxia-inducible factors, provides valuable insights into the molecular pathways driving tumorigenesis in this condition. Understanding these pathways is crucial for developing targeted therapies.

The clinical heterogeneity observed in VHL underscores the need for personalized approaches to diagnosis and management. Tailoring treatment strategies to individual patients based on their genetic and clinical profiles can optimize outcomes and quality of life.

Advances in the understanding of VHL disease at the genetic and molecular levels hold promise for the development of more effective and personalized therapeutic interventions. Targeted therapies that address the specific molecular alterations in VHL-associated tumors could revolutionize treatment outcomes.

Future Directions: Continued research into the genetics, pathophysiology, and clinical management of VHL disease is essential for improving patient outcomes and quality of life. Collaborative efforts across disciplines, including genetics, oncology, and molecular biology, will be crucial for advancing our understanding of VHL and translating this knowledge into innovative therapeutic approaches.

The present review delves into the intricate interplay of genetics, pathophysiology, and clinical manifestations of VHL disease. By elucidating the role of the VHL tumor suppressor gene, the dysregulation of the hypoxia-inducible factor (HIF) pathway, and the diverse clinical presentations, this work provides a comprehensive overview of the disease. While significant strides have been made in understanding the underlying mechanisms of VHL disease, challenges persist in developing effective therapies for all tumor types. Furthermore, the influence of regional factors, such as founder effects, genetic background, environmental exposures, and healthcare disparities, underscores the need for specific approaches to prevention, early detection, and treatment. Continued research is imperative to unravel the complexities of VHL disease, identify novel therapeutic targets, and improve the quality of life for affected individuals. A multidisciplinary approach, encompassing genetics, pathology, clinical research, and patient care, is essential to advancing our understanding and management of this debilitating condition ultimately improving outcomes for individuals affected by this complex genetic syndrome.

## Figures and Tables

**Figure 1 diagnostics-14-01909-f001:**
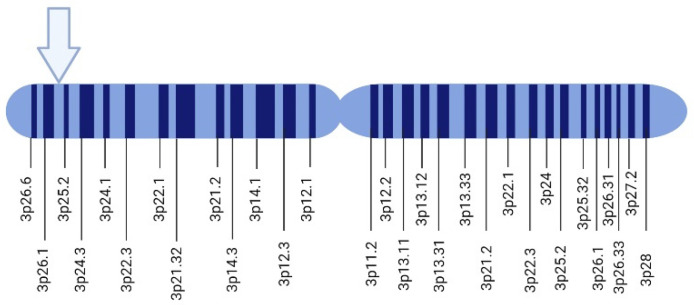
Localization of *VHL* gene in chromosome 3. VHL gene site is indicated with an arrow. Created with BioRender.com. https://app.biorender.com/illustrations/615d1c9a424d8e00af83a523 (accessed on 5 October 2021).

**Figure 2 diagnostics-14-01909-f002:**
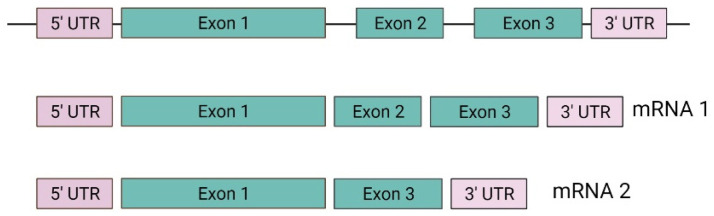
Structure of *VHL* gene. Created with BioRender.com. https://app.biorender.com/illustrations/618abf25ed4ee4026e2eb7a7 (accessed on 5 October 2021).

**Figure 3 diagnostics-14-01909-f003:**
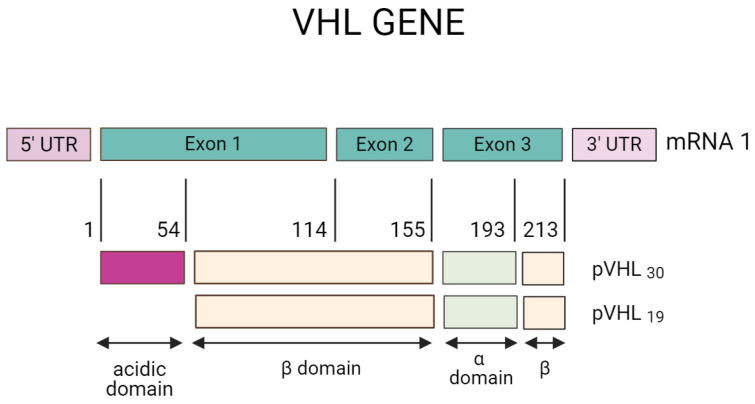
Structure of pVHL protein with two domains: the β domain that ranges from amino acids 63 to 155 and the α domain that includes amino acids 156 to 193. Created with BioRender.com. https://app.biorender.com/illustrations/618abf26ed4ee4026e2eb7ae (accessed on 5 October 2021).

**Figure 4 diagnostics-14-01909-f004:**
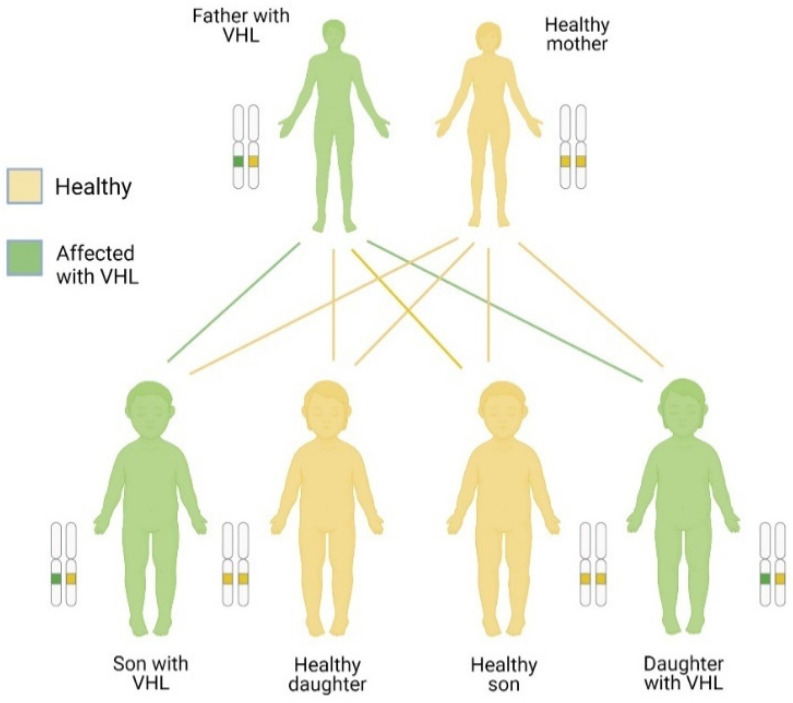
Inheritance of a dominant gene. Created with BioRender.com. https://app.biorender.com/illustrations/618abf25ed4ee4026e2eb799 (accessed on 5 October 2021).

**Figure 5 diagnostics-14-01909-f005:**
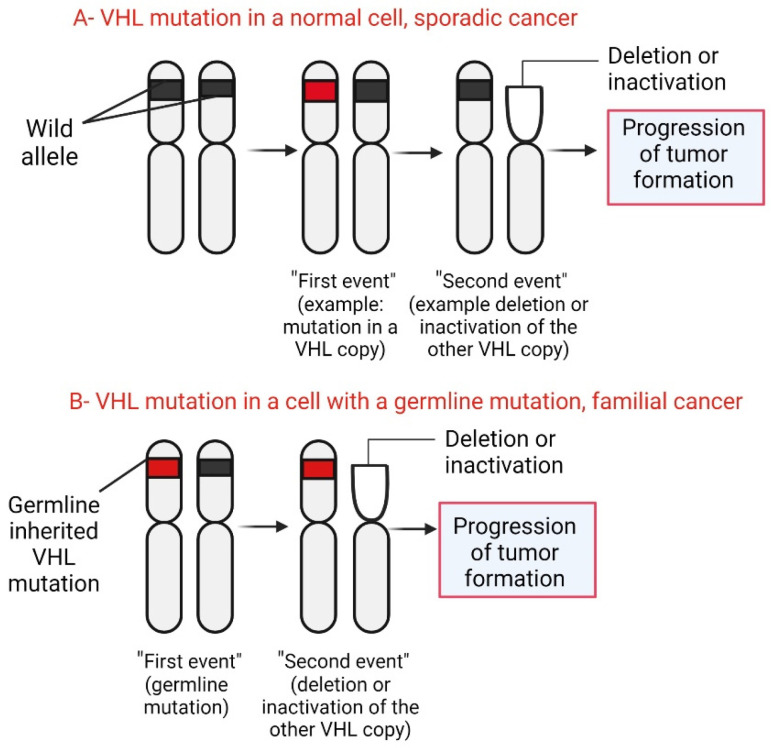
Scheme explaining tumor formation in VHL. Created with BioRender.com. https://app.biorender.com/illustrations/615d1c396f426500a826a0e7 (accessed on 5 October 2021).

**Figure 6 diagnostics-14-01909-f006:**
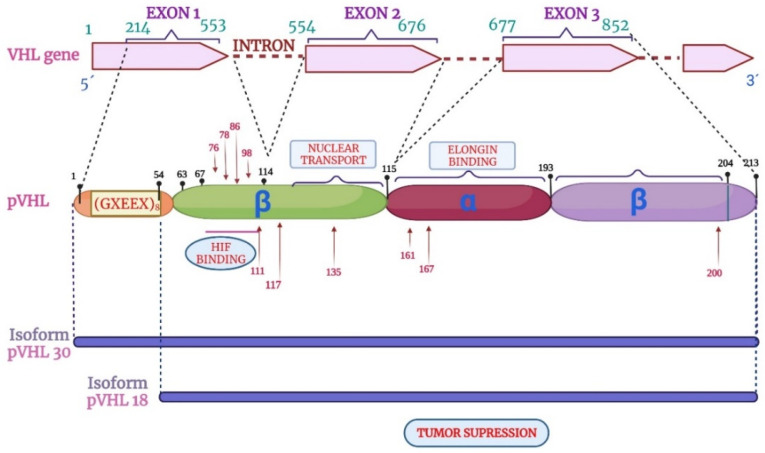
Major pVHL mutations associated with development of tumorigenesis in VHL disease. VHL protein has a beta, alpha, beta domain. Two isoforms are known according to the molecular weight of pVHL30 and pVHL18/19, and both proteins are tumor suppressors. Created with BioRender.com. https://app.biorender.com/illustrations/61eb579664332600a3d2a456 (accessed on 5 October 2021).

**Figure 7 diagnostics-14-01909-f007:**
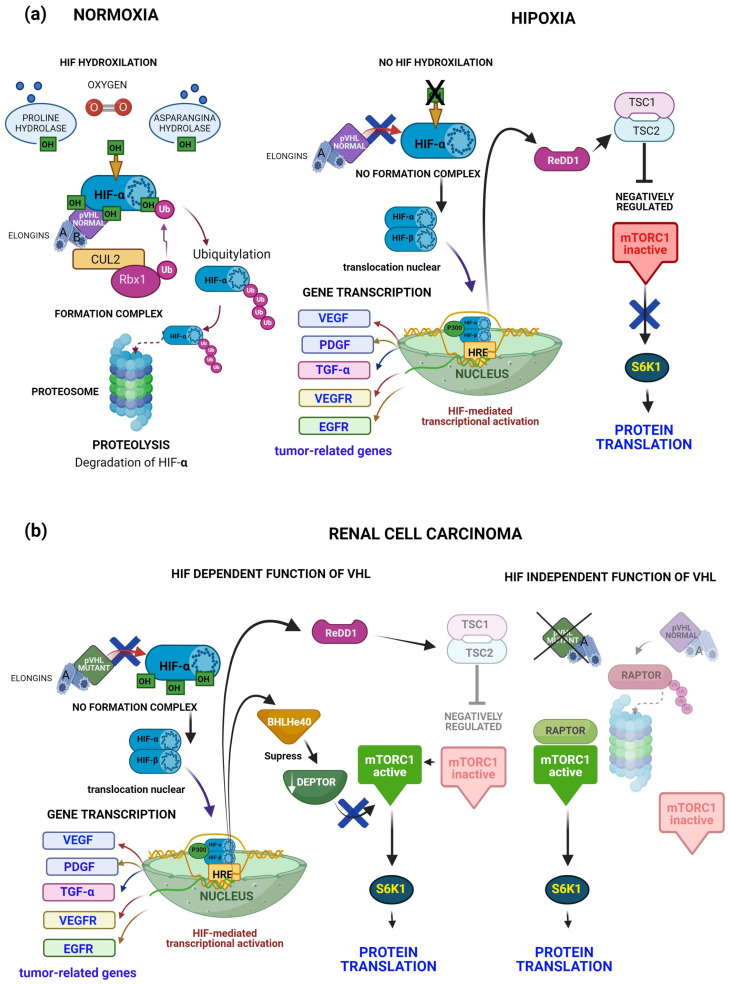
pVHL function under different conditions. (**a**) Normoxic and hypoxic conditions. Under normoxic conditions, pVHL interacts with hydroxylated HIF-α. HIF-α is ubiquitylated and degraded via the proteasome. In hypoxic conditions, pVHL does not recognize HIF-α. Then, HIF-α dimerizes with HIF-β. The complex translocates to the nucleus and promotes the transcription of many genes such as *VEGF*, *PDGF*, *TGF-α*, *EGFR*, and *VEGFR*. Another gene activated by HIF-α is ReDD1 and negatively regulates mTORC1 through Tuberous Sclerosis Complex 2 (TSC2). (**b**) In renal cell carcinoma conditions, the pVHL mutant does not bind to HIF-α and promotes the transcription of genes involved in tumor progression. Another gene up-regulated by HIF is ReDD1, but mTORC1 is activated. Thus, another HIF-dependent mechanism that activates mTORC1 is DEPTOR down-regulation. However, pVHL also regulates the deregulation of mTORC1 through the repression of RAPTOR in renal cell carcinoma. RAPTOR activates mTORC1 by destroying pVHL function. (**a**) https://app.biorender.com/illustrations/6148d17a83462b00a432cd7a (accessed on 28 January 2022); (**b**) https://app.biorender.com/illustrations/6149f556e527f3009ff8457f (accessed on 28 January 2022).

**Figure 8 diagnostics-14-01909-f008:**
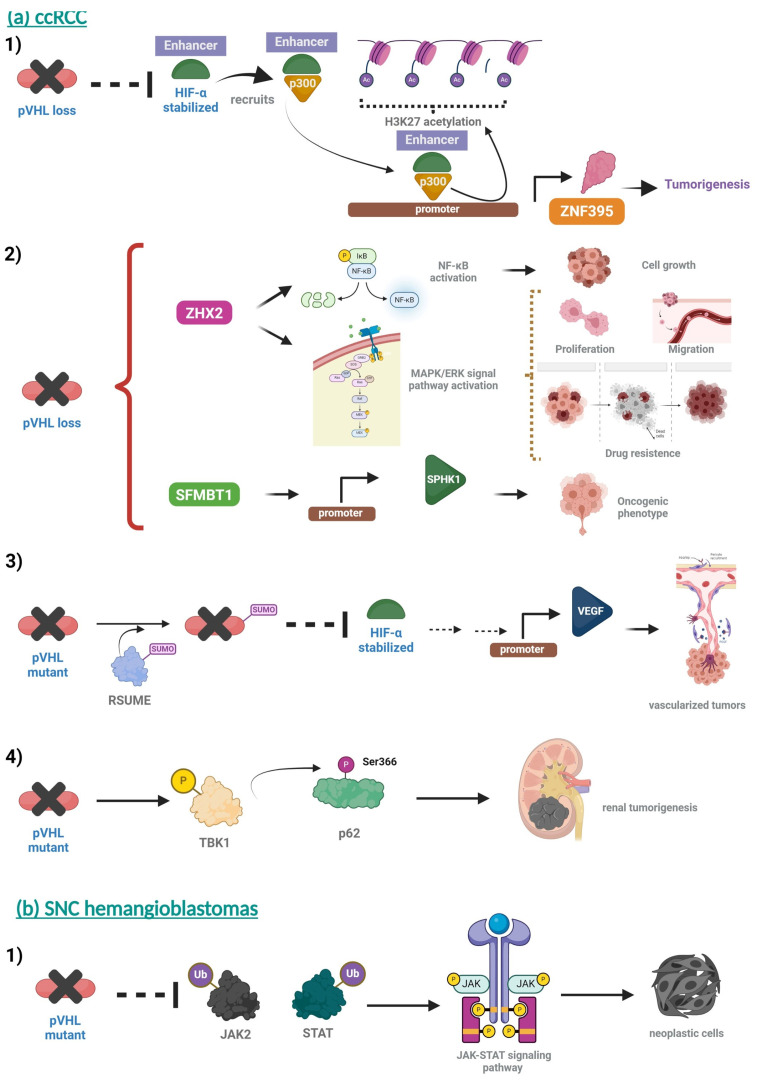
Other targets of pVHL in ccRCC and SNC hemangioblastomas. (**a**) ccRCC. (1) Loss of pVHL stabilizes HIF-α at enhancers, recruits p300 to maintain H3K27 acetylation, and up-regulates ZNF395. ZNF395 promotes tumorigenesis of ccRCC. (2) ZHX2 and SFMBT1 are novel pVHL substrates. ZHX2 promotes NF-kB activation, and thus cell growth in ccRCC. On the other hand, ZHX2 also activates the MAPK/ERK signal pathway, facilitating several cell functions in addition to causing drug resistance. SFMBT1 promotes SPHK1 expression that contributes to the oncogenic phenotype. (3) RSUME sumoylates pVHL mutants. This post-translational modification stabilizes HIF-α, and thus enhances VEGF action promoting vascularized tumors. (4) Loss of pVHL hyperactivates TBK1 phosphorylating p62 and promotes renal tumorigenesis. (**b**) SNC hemangioblastomas. (1) The pVHL mutant does not ubiquitinate STAT and JAK2. It activates the JAK-STAT signaling pathway, thus differentiating hemangioblast progenitor cells into neoplastic cells. (**a**) https://app.biorender.com/illustrations/6153323d222f2b00a627ee06 (accessed on 28 January 2022); (**b**) https://app.biorender.com/illustrations/61538464cf0d5b009562fb32 (accessed on 28 January 2022).

**Table 1 diagnostics-14-01909-t001:** Clinical classification of VHL and associated mutations. Information obtained from VHLdb database (http://vhldb.bio.unipd.it/mutations, accessed on 26 August 2024) according to [52]. Database was revised in 30 September 2021.

Clinical Classification	Associated Mutations	References
von Hippel–Lindau syndrome Type 1	p.Asn78Ser	[53]
	p.Pro81Ser	[54]
	p.Trp117Cys	[55]
	p.Phe136Ser	[53]
	In-frame del	[56]
	Stop at 158	[57]
	p.Asn78His	[58]
	p.Ser65Trp	[59]
	p.Leu89Pro	[60]
	p.Ser111Asn	[61]
	p.Ser111Arg	[62]
	p.Ser80Arg	[63]
	p.Pro86Leu	[55]
	p.Ser65Leu	[61]
	p.Trp88Arg	[64]
	In-frame del Phe76	[54]
	p.Gln96Pro	[65]
	p.Trp88Ser	[54]
	p.Gln73X	[66]
	p.Asn90Ile	[67]
	p.Leu184Pro	[54]
	p.Ile180Val	[60]
	p.Cys162Arg	[54]
von Hippel–Lindau syndrome Type 2	p.Phe119Leu	[68]
von Hippel–Lindau syndrome Type 1 and Type 2	Frameshift	[53,69]
	p.Leu118Pro	[53,65]
	p.Leu178Pro	[54,55]
	p.Ser80Asn	[60,70]
	p.Gly93Asp	[55,71]
	p.Leu158Pro	[54,72]
	p.His115Gln	[54,73]
	p.Gly114Cys	[53,58]
von Hippel–Lindau syndrome Type 1 and Type 2B	p.Arg113X	[59,71]
von Hippel–Lindau syndrome Type 1, Type 2, and Type 2A	p.Tyr98His	[55,74]
von Hippel–Lindau syndrome Type 1, Type 2, and Type 2B	p.Arg161X	[59,75,76]
	p.Cys162Tyr	[71,77]
	p.Val74Gly	[54,74]
	p.Cys162Trp	[54,59,73]
von Hippel–Lindau syndrome Type 1, Type 2, and Type 2C	p.Leu188Val	[59,74,78]
	p.Gly93Ser	[54,70,79]
von Hippel–Lindau syndrome Type 1, Type 2A, and Type 2B	p.Pro86Ser	[65]
von Hippel–Lindau syndrome Type 1, Type 2, Type 2A, and Type 2B	p.Arg167Gln	[70,73,74,80]
	p.Arg167Trp	[59,80,81,82]
von Hippel–Lindau syndrome Type 1, Type 2, Type 2A, Type 2B, and Type 2C	p.Arg167Trp	[54,59,73,83]
von Hippel–Lindau syndrome Type 2 and Type 2A	p.Val166Phe	[73,84]
	p.Arg161Gln	[73,85]
	p.Tyr98His	[70,74]
	p.Tyr112His	[55,86]
von Hippel–Lindau syndrome Type 2 and Type 2B	p.Arg161Gly	[74,87]
von Hippel–Lindau syndrome Type 2, Type 2A, and Type 2B	p.Thr157Ile	[54,71,73]
von Hippel–Lindau syndrome Type 2A and Type 2B	p.Ala149Ser	[88,89]

**Table 2 diagnostics-14-01909-t002:** Summary of the lesions, symptoms, diagnosis, imaging, and treatment of VHL patients.

Lesion/Frequency in Patients by Age	Symptoms	Diagnosis(MR, Magnetic Resonance; CT, Computed Tomography)	Imagen	Treatment and Management	Refs.
CNS hemangioblastomas (HBs) cerebellum and spinal cordEarly third decade (ages 22–26 years). Hemangioblastomas occur in approximately 60 to 80% of patients with VHL.	Headache, gait imbalance, ataxia, abnormal head position, nausea, vomiting, and papilledema	MR—brain	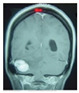 [99] http://creativecommons.org/licenses/by-nc-nd/3.0/ (accessed on 22 August 2024).	Followed by repeated MRI scans in asymptomatic patients. In patients, symptomatic tumors should be surgically removed.	[99,100]
Neurological impairment, urinary or bowel abnormalities, singultus, dysphagia, myelopathic disorders, syringomyelia, and polyglobulia.	MR—spinal cord	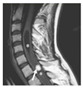 [101] http://creativecommons.org/licenses/by-nc-nd/2.0/ (accessed on 22 August 2024).	Depending on the size of the tumor, surgical removal is recommended.	[100,101]
Retinal hemangioblastoma (RH) Median age 21 and 25 years old, with a frequency from 49% to 85%.	Gradual loss of vision	Angiography Ultrasonography	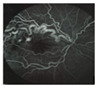 [102] http://creativecommons.org/licenses/by/4.0/ (accessed on 22 August 2024).	Ablative treatment: thermal laser photocoagulation, cryotherapy, radiation, and transpupillary thermotherapy.	[5,102,103]
Renal cystsCommonly occurs in the fourth decade of life, but this variant of VHL disease could occurrs as young as 16 years old. 60–70% of the lesions were carcinomas, all with clear cell features (RCCs, Renal Cysts Clear Cells).	Mostly asymptomatic. flank pain or hematuria	Ultrasound, Abdominal MRI, or CT	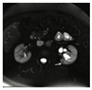 [14] http://creativecommons.org/licenses/by/4.0/.	Partial nephrectomy is the option for tumors that have reached 3 cm, reducing the risk of metastasis while maintaining kidney function. VHL RCCs were treated with radical nephrectomies.	[5,14,91,92,101]
Pheochromocytoma (PCC)Diagnosticated arround the third decade of life with a frecuency of 33%.	Headache, sweating, palpitation, and hypertension	CT (computational Tomography), MRI (Magnetic Resonance Image)	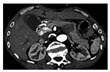 [104] http://creativecommons.org/licenses/by-nc-nd/4.0/.	Initial treatment is with alpha adrenergic blockers such as phenoxybenzamine for the control of hypertension. However, surgical resection remains the definitive treatment.	[5,44,101,104,105]
Pancreatic cysts and pancreatic neuroendocrine tumors (NETs) If diagnosis already in fourth decade, 35–75% of patients with VHL disease will develop simple pancreatic cysts.	Jaundice, abdominal pain, pruritis, vomiting and abdominal swelling	MRI, CT	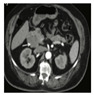 [14] http://creativecommons.org/licenses/by/4.0/.	Pancreatic cysts do not require surgical intervention; PNETs with a potential for metastatic disease are resected with enucleation by Whipple’s procedure or partial pancreatectomy depending on location and size greater than 3 cm.	[14,101,106,107,108]
Endolymphatic sac tumors (ELTS) 10 to 15% of patients with VHL disease develop ELSTs. The mean age of onset is 22 years old (range, 12–50 years) and they may be bilateral in 30% of cases.	Facial nerve palsy, vestibulocochlear impairments,tinnitus, vertigo, disequilibrium, and hearing loss	CT	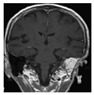 [109] http://creativecommons.org/licenses/by-nc-nd/3.0/.	Treatment of ELTS requires extensive surgery with adequate bone removal around the area of the macroscopically evident tumor.	[109,110]
Epididymal cystadenomas 16–80 years old. Occur in 25–60% of affected men	Mostly asymptomatic	UltrasonographyPET/CT images	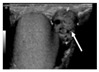 [111] http://creativecommons.org/licenses/by/4.0/.	Surgery is rarely performed in epididymal cystadenomas. Routinely followed by a physical exam and ultrasonography.	[92,111,112]
Broad ligament cystadenomasDiagnosticated about the second decade with 10% frecuency.Women unilateral presentation.	Mostly asymptomatic	Transvaginal ultrasound	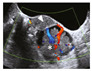 [113] http://creativecommons.org/licenses/by/4.0/.	Being benign lesions, they are usually managed conservatively without surgery.	[113,114]

**Table 3 diagnostics-14-01909-t003:** Animal models for von Hippel–Lindau disease.

Organ System	Model Organism	Phenotype	Limitations
Central Nervous System (CNS)	Zebrafish	Retinal neovascularization, edema, retinal detachment	No hemangioblastoma development
Kidney	Mouse	Renal microcysts, macrocysts, acute nephritis, hematuria, proteinuria, renal failure	Limited to specific kidney regions
Pancreas	Mouse	Highly vascularized cysts, microcystic adenomas	Incomplete penetrance, limited to pancreas
Reproductive System	Mouse	Clear cell cystadenoma of genital tract	Specific to genital tract epithelium
Liver	Mouse	Hepatic hemangiomas, angiectasias, steatosis, inflammatory cell infiltration	Embryonic lethality in some models, limited to liver and other specific organs

**Table 4 diagnostics-14-01909-t004:** Biomarkers in von Hippel–Lindau disease.

Biomarker	Sample	Cohort	Results/Remarks	Ref.
Telomere length	Blood	300 VHL patients 92 healthy family controls	▪ Patients with VHL showed significantly shorter telomere lengths than healthy family controls. ▪ Patients in the shorter telomere group suffered higher age-related risks of VHL-associated central nervous system hemangioblastomas, renal cell carcinoma, pancreatic cyst, and neuroendocrine tumors. ▪ The results indicate that shorter blood telomere length is a biomarker of risk in VHL patients.	[156]
c-Myc/HIF-2α	Formalin-fixed, paraffin-embedded primary tumor samples	69 patients with metastatic renal cell carcinoma (mRCC), a component of clear cell histology, and sunitinib as first-line treatment.	▪ Double c-Myc/HIF-2α-positive staining tumors showed a significant association with a lower PFS and a significantly worst response to sunitinib in patients with mRCC.	[157]
Serpin peptidase inhibitor clade H member 1 (SERPINH1)	Primary ccRCC and adjacent normal kidney tissues	Four stages of ccRCC and adjacent normal tissues of 18 patients.	▪ Multivariate Cox analysis revealed that SERPINH1 is a possible prognostic marker for DFS in patients with VHL-WT ccRCC.	[158]
Programmed death ligand 1 (PD-L1)	Formalin-fixed and paraffin-embedded surgical RCC simples	26 patients with VHL disease with ccRCC. 129 patients with sporadic ccRCC.	▪ Positive PD-L1 in ccRCC was associated with aggressive clinicopathological characteristics both in sporadic and VHL-associated hereditary ccRCC patients.	[159]
Lamin B1	Sample tissues (primary tumors and normal tissues samples)	932 patients with primary RCCs.	▪ This study identified high expression of LMNB1 as an unfavorable prognostic marker in ccRCC patients.	[160]
Cytotoxic T-Lymphocyte Antigen 4 (CTLA-4) and PD-1	Formalin-fixed paraffin embedded (FFPE) specimens collected from radical surgery	119 patients who underwent surgery without neoadjuvant chemotherapy and were diagnosed with kidney renal clear cell carcinoma.	▪ Patients in the high-risk group with poor survival had a positive correlation with low fractions of CD4 + T cells and dendritic cells and exhibited a lower expression of CTLA-4 and PD-1 than the low-risk group.	[161]
lncRNA FGD5-AS1	Renal tissue (adjacent normal renal tissue and ccRCC tissue)	54 patients including 12 ccRCC caused by VHL mutation.	▪ The expression of FGD5-AS1 was significantly lower in ccRCC than in adjacent normal tissues, and increased FGD5-AS1 was associated with longer OS and DFS.	[162]
SFMBT1 and ZHX2	Tumor tissues and adjacent normal tissues	97 patients with ccRCC.	▪ High levels of SFBMT1 and ZHX2 expression were associated with poor clinical outcomes in patients with ccRCC.	[135]
^89^Zr-bevacizumab	Imaging	22 patients with at least 1 measurable hemangioblastoma were eligible.	▪ ^89^Zr-bevacizumab PET can visualize different manifestations of VHL disease but does not predict the behavior of a lesion, but could predict sensitivity to antiangiogenic treatment.	[163]
^68^Gallium-DOTATATE PET/CT	Imaging	36 patients with VHL diagnosis associated to pancreatic lesions.	▪ ^68^Ga-DOTATATE PET/CT had a significantly higher detection rate compared with CT and MRI.	[164]

**Table 5 diagnostics-14-01909-t005:** Clinical trials for von Hippel–Lindau disease.

Clinical Trial Identifier and Therapeutic Agent	Clinical Phase	Status	Condition or Disease	Molecular Target	Participants	Outcome	Refs.
NCT00673816 Sunitinib	II	Terminated	Advanced Ocular Disease of von Hippel–Lindau Syndrome	Inhibition of multiple receptor tyrosine kinases (RTK), including the vascular endothelial growth factor (VEGF) and platelet-derived growth factor (PDGF)	5 patients receive 9 months of sunitinib malate therapy administered in 6 cycles. Each cycle consisted of a daily oral dose of 50 mg sunitinib malate for 4 weeks followed by a 2-week rest period. (Only one completed treatment).	Change in Best-Correction Visual Acuity (BCVA), Retinal Thickness, and Retinal Angioma Leakage from Baseline to Week 36. The recruitment goal was to enroll five participants; however, the study was terminated after only two participants had been enrolled due to slow recruitment and adverse events.	[169]
NCT00330564 Sunitinib	II	Terminated	von Hippel–Lindau syndrome: Renal cell carcinoma and/or Hemangioblastoma		Treatment with SU011248/sunitinib malate (50 mg daily dose for 4 weeks, then 2 weeks off) for 6 months in 15 patients with von Hippel–Lindau (VHL) syndrome who have a measurable lesion undergoing surveillance.	Nine of the fifteen patients completed all four cycles of therapy, and the expected toxic effects were responsible for the necessary dosage reductions and discontinuation of treatment. Renal cell carcinomas responded better to sunitinib therapy than other VHL related lesions using the RECIST measure. The study ended early due to slow accrual.	[170]
NCT02859441 Ranibizumab E10030	I/II	Completed	von Hippel–Lindau (VHL) Retinal Capillary Hemangiomas (RCH)	E10030, a PDGF-B antagonist, and ranibizumab, a VEGF-A antagonist	This was a single-arm open-label phase 1/2 study, consisting of 3 adults with VHL-associated RH and vision loss. Intravitreous injections of ranibizumab (0.5 mg) and E10030 (1.5 mg) were administered unilaterally and each received 9 injections prior to week 52 and were followed for 104 weeks.	One participant manifested mild episodic ocular hypertension in the study eye. The change in BCVA in the study eye at week 52 for the three participants was –5, –12, and +2 letters. No reduction in RH size was measured at 52 weeks. Variable mild improvements in exudation in two participants at week 16 were not sustained through week 52. Intranavitreous injection with ranibizumab and E10030 demonstrated a reasonable preliminary safety profile, but limited treatment effect.	[171]
Protocol number CT/AE-941/002 Neovastat (Canadá)	II	Completed	Renal cell carcinoma (RCC)	Inhibition of angiogenesis	22 patients with a primary diagnosis of refractory CCR. They were treated with Neovastat 240 mL/day (*n* = 14) compared to patients receiving 60 mL/day (*n* = 8).	The higher dose of Neovastat administered in this trial is associated with a survival benefit in RCC. Neovastat is well tolerated by advanced cancer patients at doses of 60 and 240 mL/day.	[172]
Not definite. Neovastat (Canadá)	III	Completed	Metastatic renal cell carcinoma in whom immunotherapy failed		300 patients from 48 international centers were randomized to receive 120 mL twice daily of oral Neovastat or placebo Neovastat. The 300 patients who received at least 1 dose of study medication were included in this analysis.	The study of metastatic CRC provides a prognostic model that has a significant impact on risk-adjusted survival. Although external validation in an independent data set is lacking, the results of this trial may lead to a new paradigm for clinical trial design and risk stratification when considering future investigations of patients with metastatic CRC in whom immunotherapy has failed.	[173]
EudraCT Number: 2014-003671-30 Propanolol (Spain)	III	Completed	von Hippel–Lindau disease and retinal hemangioblastomas	VEGF inhibitor	7 patients were included. All patients received a daily dose of 120 mg propranolol for 1 year. Clinical variables were evaluated at baseline, and at 1, 3, 6, 9, and 12 months.	The number and size of retinal hemangioblastomas remained stable in all patients. The only adverse effect reported was hypotension in one patient. The results suggest that propranolol could be useful for the treatment of retinal hemangioblastomas in patients with VHL, especially when there are retinal exudates. The results of this clinical trial allowed propranolol designation to treat von Hippel–Lindau disease, granted by the European Medicines Agency (EMA).	[174]
EudraCT Number: 2007-002132-29 Sorafenib (UK)	II	Terminated	VHL-associated renal cancer	Tyrosine kinase inhibitor	4 patients with VHL syndrome who had therapy for advanced RCC, received sorafenib orally (400 mg twice a day) for up to six months.	This study concludes that over a 6-month period of sorafenib, at the standard dose used in RCC, there was no response effect in CNS hemangioblastomas in this population of patients.	[175]
NCT01436227 Pazopanib	II	Active, not recruiting	von Hippel–Lindau disease genetically confirmed or one disease-related lesion.	Vascular endothelial growth factor receptors (VEGFR) -1, -2, and -3, c-kit and platelet-derived growth factor receptor (PDGF-R) inhibitors	31 eligible patients were treated with pazopanib 800 mg by mouth daily for 24 weeks.	To ensure timely dissemination of data, the decision was made to close the trial after 31 evaluable patients were accrued. Pazopanib induces a reduction in the burden of the disease in von Hippel–Lindau disease patients. Efficacy data indicate benefit in individuals with renal cell carcinomas and pancreatic lesions, and some potential efficacy signals in hemangioblastomas as well. Pazopanib could be considered in patients with von Hippel–Lindau disease and growing lesions where surgical resection may be required in the relatively near future, or in patients with unresectable lesions where a decrease in tumor size is desired.	[176]
NCT03401788 Belzutifan	II	Active, not recruiting	von Hippel–Lindau Disease-associated renal cell Carcinoma	Inhibitor of HIF-2α	61 patients receive 120 mg of belzutifan orally once a day until progression, intolerable toxicity, or the investigator/patient’s decision to withdraw.	Of the 61 patients, 53 (86.9%) had a decrease in the size of the target lesions. Responses were also observed in CNS, retinal, and pancreatic lesions. MK-6482 showed promising efficacy and tolerability in patients with VHL-associated ccRCC and responses in other VHL-related lesions.	[177,178]
NCT03108066 PT2385	II	Active, not recruiting	von Hippel–Lindau Disease-associated renal cell carcinoma	Inhibitor of HIF-2α	4 patients were enrolled in each stage of a two-stage design. PT2385 was administered orally at a dose of 800 mg twice daily, with a follow-up of 19 weeks.	All patients had stable disease (SD) as their best response at the latest assessment. PT2385 demonstrated stabilization of disease in VHL-associated clear cell RCC and other tumors, and showed an acceptable safety profile.	See below
NCT01266070 Dovitinib	II	Terminated	VHL-related hemangioblastoma	A multityrosine kinase that inhibits FGFR, VEGFR, and PDGFR.	6 participants received 500 mg/day (5 days in/2 days out of dosing). All participants completed at least two cycles of therapy.	The trial was stopped after six patients due to the activation of the toxicity stopping rule. The lack of response in HBs in this population treated with dovitinib is surprising, and molecular profiling of HB tissue would be extremely useful to help understand the biologic underpinnings of this lack of efficacy.	[179]
NCT00089765 Ranibizumab	I	Completed	Angiomas (blood vessel tumors) in patients with von Hippel–Lindau syndrome (VHL)	VEGF-neutralizing agent	5 patients with retinal capillary hemangioblastomas (RCH) associated with VHL with exudative changes and visual loss. Monthly intravitreal injections of ranibizumab (0.5 mg) were administered over a 6 month course for a total of 7 injections, with additional injections considered until week 52.	The primary outcome was the change in best-corrected visual acuity (BCVA). Secondary outcomes included change in lesion size, change in retinal thickness, and adverse event assessments. Intravitreal ranibizumab, administered as monotherapy every 4 weeks, had minimal beneficial effects on most RCHs related to VHL. Future studies are needed to determine a combination with other therapies for the treatment of ocular tumors associated with VHL.	[180]
NCT02108002 Vorinostat	I	Completed	VHL-related hemangioblastoma	Histone deacetylase inhibitor (HDACi)	7 germline missense VHL patients with symptomatic CNS hemangioblastomas received 400 mg of vorinostat by mouth daily for seven days prior to surgery and subsequently underwent surgical resection.	Vorinostat is well tolerated by patients with symptomatic CNS hemangioblastomas in the context of germline missense VHL disease and shows results in mutated stabilization of the pVHL protein. This suggests that vorinostat may be a promising treatment for patients with a germline mutation.	See below
NCT00566995 Vandetanib	II	Completed	VHL-associated renal cell carcinoma	Dual VEGFR2/EGFR inhibitor	34 participants received a 300 mg/day (starting dose) oral dose of vandetanib for 28 days.	Vandetanib demonstrated antitumor activity. However, the poor tolerability required drug withdrawal in a significant proportion of patients. Newer agents that selectively target VEGF receptors may offer a more tolerable alternative and could optimize clinical benefits in this population.	[181]
Medical Ethics Committee of Peking University First Hospital (Beijing, China) TKIs	ND	Completed	von Hippel–Lindau disease	Tyrosine kinase inhibitor (TKI)	32 patients receiving TKIs were recruited. For sunitinib, a dosage of 50 mg/day was administered orally for 28 days, followed by a 14-day break per cycle for several cycles. For sorafenib, a dose of 800 mg/day divided into two doses was administered orally. For axitinib, a dose of 10 mg/day divided into two doses was administered orally. For pazopanib, a dose of 800 mg/day was administered orally.	A partial response was observed in eleven (31%) of thirty-six renal cell carcinomas, four (27%) of fifteen pancreatic lesions, and one (20%) of five central nervous system (CNS) hemangioblastomas. The mean tumor size decreased significantly for renal cell carcinomas (*p* = 0.0001), renal cysts (*p* = 0.027), and pancreatic lesions (*p* = 0.003) after TKI therapy. Finally, the side effects were acceptable.	[182]

## Data Availability

Not applicable.

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
