# Peer review of "Genetics, Pathophysiology, and Current Challenges in Von Hippel–Lindau Disease Therapeutics"

_diagnostics, 2024, doi:10.3390/diagnostics14171909_

Round 1
Reviewer 1 Report
Comments and Suggestions for Authors
June 28, 2024
Ms. Ref. No.: diagnostics-3081949
Journal: Diagnostics.
Title: Genetics, Pathophysiology, and Current Challenges in Von Hippel-Lindau Disease Therapeutics.
Comments:
Thank you for your efforts in composing an on such a pertinent subject and many thanks for its valuable figures. I have taken the liberty of providing you with a few observations that I believe will serve to enhance the quality of your work. Please find my feedback outlined in the following paragraphs
1- Every words of title are begun with capital letters, please recheck them.
2- It seems to be better that write “VHL gene” in Italic style.
3- In the abstract part, the authors only told what they want search (for example: we review the etiology, epidemiology, pathophysiology, genetics, clinical manifestations, diagnosis, and current treatments, as well as the molecular aspects of this disease.) and did not tell about results please rewrite the main points and results in the abstract.
4- The first part of introduction (line 31 up 68) did not have any reference. Please recheck it.
5- Please write the complete of aa= Amino Acids in Figure 3.
6- In the Tables of this manuscript, mentioned the number of each references, is it possible that mentioned additional information about these references?
7- Is the style of Tables 2 & 3, according to this journal guideline?
8- It is better to summarize the data of section “6. Animal models for the study of VHL lines 385 up 436) in tables such as Tables 1 &2.
9- According to the results of this article, please reintroduce the effect of regional populations with this disease?
10- It is better to clarify the main results in the conclusions part of this article.
11- There are 165 valuable references in this manuscript, what was the criteria for referencing them?
12- Additionally, what was the inclusion and exclusion criteria for these references?
Author Response
Comments 1:
"Every word of title are begun with capital letters, please recheck them".
Thank you for pointing this out, we agree with this comment, and we have corrected it making each word begin with a capital letter. (Please check the revised version of the manuscript, first page lines 1 and 2).
Comments 2:
"It seems to be better that write “VHL gene” in Italic style".
Thank you for pointing this out, we agree with this comment, and we have corrected it within the text of the manuscript each time we refer to the VHL gene in italic font., please find all the corrections in the revised manuscript marked with the control of changes from Microsoft Word.
Comments 3:
"In the abstract part, the authors only told what they want search (for example: we review the etiology, epidemiology, pathophysiology, genetics, clinical manifestations, diagnosis, and current treatments, as well as the molecular aspects of this disease.) and did not tell about results please rewrite the main points and results in the abstract".
Thank you for pointing this out, we agree with this comment, and we have rewritten the abstract including the results of our search highlighting the main points as you suggested. (First page, first paragraph, lines 18-40, of the revised manuscript).
Comments 4:
"The first part of the introduction (line 31 up 68) did not have any reference. Please recheck it".
Thank you for pointing this out, we agree with this comment, and we have Rewrite the introduction and included the corresponding references. (Page 2, lines 56-81).
Comments 5:
"Please write the complete of aa= Amino Acids in Figure 3".
Thank you for pointing this out, we agree with this comment, and we have corrected it, each time we use the abbreviation aa, we have changed for the complete word: Amino Acids (in the legend of Figure 3 and the first paragraph of page 5, lines 222-224. and page 6 line 253).
Comments 6:
"In the Tables of this manuscript, mentioned the number of each references, is it possible that mentioned additional information about these references?"
Thank you for pointing this out, we agree with this comment, and we have included a paragraph before tables 4 (this is the new number of the table, since we introduced a new table) and 5 (this is the new number of the table, since we introduced a new table) summarizing the relevant information about the references cited in these tables. (Page 20, Line 553-571 and Page 23 Line 633-652).
Comments 7:
"Is the style of Tables 2 & 3, according to this journal guideline?"
Thank you for the comment, we have adjusted the style of our tables to the Journal guidelines.
Comments 8:
"It is better to summarize the data of section “6. Animal models for the study of VHL lines 385 up436) in tables such as Tables 1 &2".
Thank you for pointing this out, we agree with this comment, and we have included a new table summarizing the information about Animal models for the study of VHL in this section. (Page 19, Line 522).
Comments 9:
"According to the results of this article, please reintroduce the effect of regional populations with this disease?"
Thank you for pointing this out, we agree with this comment, and we have included a new section in the manuscript entitled: Effect of Regional Populations on VHL Disease. (Page 22, Lane 607-630).
Comments 10:
"It is better to clarify the main results in the conclusions part of this article".
Thank you for pointing this out, we agree with this comment, and we have included a final paragraph at the end of the conclusion section that summarizes the main results of our review (Line 752-765)
Comments 11:
"There are 165 valuable references in this manuscript, what were the criteria for referencing them?"
Thank you for your comment, We selected those 165 references according to their relevance to our review of VHL, we considered all of them important and relevant to our review paper after doing a wide references search in this field of study.
Comments 12:
"Additionally, what was the inclusion and exclusion criteria for these references?"
We apply these criteria consistently, to ensure that our VHL review manuscript provides a comprehensive, up-to-date, and reliable overview of the research field:
Inclusion Criteria:
- Relevance to the topic: The reference should directly contribute to the central theme or research question of the review.
- Quality of evidence: Prioritize studies with robust methodologies, such as randomized controlled trials, meta-analyses, or systematic reviews.
- Currency: Include recent studies to reflect the current state of knowledge, but also consider landmark or foundational studies.
- Diversity: Incorporate a variety of sources, including primary research articles, review articles, and potentially books or book chapters.
- Comprehensiveness: Aim for a balanced representation of different perspectives and methodologies within the field.
Exclusion Criteria:
- Irrelevance: References that do not directly relate to the review's focus should be excluded.
- Low quality: Studies with methodological flaws or limitations may be excluded, especially if there are higher-quality alternatives.
- Outdated information: Very old studies may be excluded if more recent research provides better evidence, except for the original first publications in the field.
- Limited accessibility: References that are difficult to obtain or verify may be excluded if there are sufficient alternatives.
Please see the attachment with the file of the reviewed manuscript with all the corrections.

Reviewer 2 Report
Comments and Suggestions for Authors
The authors provided a very detailed review of Von Hippel-Lindau (VHL) disease, a rare autosomal dominant genetic disorder with multiple organ neoplastic syndrome. See the comments in the attached pdf file. I suggested an option of including potential links between HIF activation and tumor inhibition in VHL to the PI3K/Akt signaling pathway.

Author Response
Comment 1:
"The authors provided a very detailed review of Von Hippel-Lindau (VHL) disease, a rare autosomal dominant genetic disorder with multiple organ neoplastic syndrome. See the comments in the attached pdf file. I suggested an option of including potential links between HIF activation and tumor inhibition in VHL to the PI3K/Akt signaling pathway."
Thank you for your valuable suggestion to improve our paper. We have included a paragraph in the section of Pathophysiology, about potential links between HIF activation and tumor inhibition in VHL to the PI3K/Akt signaling pathway. And we have cited the papers that you suggested: https://doi.org/10.1038/s41523-023-00598-z and https://doi.org/10.3389/fonc.2014.00064 (Page 4 Line 175 and Page 13 Line 360).
We have also made all the corrections that you suggested in the PDF of our manuscript.
Please see the attachment with the Reviewed version of our manuscript with all the corrections.

Round 2
Reviewer 1 Report
Comments and Suggestions for Authors
Many thanks for the revision file that was sent.
According to the author's response, the authors answered my question well and made the necessary corrections.
An additional issue that appeared in the revision profile of this manuscript is about iThenticate after revision. It is about 42 percent match.
It seems to be better to recheck this percentage due to journal guidelines.